# The Effect of Reduced Crude Protein on Growth Performance, Nutrient Digestibility, and Meat Quality in Weaning to Finishing Pigs

**DOI:** 10.3390/ani13121938

**Published:** 2023-06-09

**Authors:** Young-Geol Han, Geon-Il Lee, Sung-Ho Do, Jae-Cheol Jang, Yoo-Yong Kim

**Affiliations:** 1Department of Agricultural Biotechnology and Research Institute of Agriculture and Life Sciences, Seoul National University, 1, Gwanak-ro, Gwanak-gu, Seoul 08826, Republic of Korea; hanyounggur@naver.com (Y.-G.H.); thel6101@naver.com (S.-H.D.); 2Department of Animal Science, Chonnam National University, 134, Uchi-ro, Buk-gu, Gwangju 61186, Republic of Korea; clerk123@naver.com; 3Division of Animal Science, College of Agricultural Life Sciences, Gyeong Sang National University, 33 Dongjin-ro, Gyeonsangnam-do, Jinjusi 52725, Republic of Korea; jaejang1278@gnu.ac.kr

**Keywords:** crude protein, low-protein, high-protein, diarrhea, fatty acids, growing pigs

## Abstract

**Simple Summary:**

Nutrient waste management poses significant challenges associated with environmental concerns and feed expenses for the pig industry. Additionally, including high levels of crude protein in pig diets has been linked to postweaning diarrhea and protein fermentation in the large intestine compared to diets with a lower protein content, which can impair gut health. This study was conducted to assess the impact of low-protein diets on weaning to finishing pigs related to growth performance, nutrient digestibility, and meat quality. Our findings indicate that when supplemented with appropriate amino acids, low-protein diets do not negatively affect growth performance or nutrient digestibility in weaning to finishing pigs. Furthermore, the quality was superior in pigs fed low-protein diets as opposed to those fed high-protein diets. Consequently, employing low-crude-protein diets in weaning to finishing pigs is recommended.

**Abstract:**

This study aimed to evaluate optimal crude protein (CP) levels based on the National Research Council guidelines from 1998 and 2012 and their impacts on growth performance, fecal score, blood profiles, nutrient digestibility, and carcass characteristics of weaning to finishing pigs. Four diets were established in this experiment: high-protein (HP), medium-high-protein (MHP), medium-low-protein (MLP), and low-protein (LP). The HP diet followed the NRC (1998) guidelines, while the MHP diet reduced the CP content by 1%. The MLP diet had 1.5% lower CP content than the HP diet. The LP diet followed the NRC guideline of 2012, which suggests less protein than the NRC guideline of 1998. There were no significant differences in body weight, average daily feed intake, or nutrient digestibility. However, the average daily gain (ADG) of pigs fed the LP diet at 7–10 weeks was lower than in the other treatments, whereas the ADG of the pigs fed the LP diet was higher compared with that of pigs fed the other diets, showing compensatory growth in finishing periods (*p* < 0.05). The blood urea nitrogen of pigs fed the LP diet showed the lowest value, whereas the highest value was found in pigs fed the HP diet, and the other two diets were similar to the HP diet or positioned between the HP and LP diets (*p* < 0.05). Water holding capacity, cooking loss, shear force, and pH in the longissimus muscle were not influenced by varying dietary CP levels (*p* > 0.05), but the hunter values L and b were increased in pigs fed the LP diet (*p* < 0.05). Conclusively, a low-protein diet did not negatively affect growth performance, nutrient digestibility, or meat quality.

## 1. Introduction

During the last several decades, the dissipation of energy and nutrients, especially nitrogen and carbon, has become a primary concern from the standpoint of environmental regeneration. Researchers in the feed industry have endeavored to reduce nitrogen waste to minimize its environmental impact, and thus a low dietary crude protein (CP) content in feedstuffs has been suggested to decrease the negative influence of high-protein (HP) diets, such as increased nitrogen excretion and feed cost [1,2,3,4]. The impact of a high- or low-protein diet can vary depending on the farm environment, the growth development of pigs, and pig conditions. Therefore, reducing CP content through the supplementation of amino acids (AA) has been recommended to improve feed efficiency, taking into account different environmental factors.

Despite meeting AA requirements for lysine and methionine at specific levels, the HP diet commonly used in weaning and growing pigs has been associated with several issues, including the development of unhealthy gut flora, diarrhea, and intestinal disease within the gastrointestinal tract [5,6,7]. Undigested dietary protein in pigs can be directed to the cecum, where microbiota can use it to undergo protein fermentation in the large intestine [8]. However, this process has been linked to the production of toxic metabolites (sulfuric compounds, indole, phenolic compounds, amines, and ammonia), which may inhibit normal microbial activity and hinder the proliferation of epithelial cells within the large intestine [9]. The presence of these harmful compounds can result in odor pollution. Therefore, the reduction of CP levels in pig diets is an inevitable necessity.

Diverse nutritional approaches have been suggested to reduce CP content in pig feed [10,11,12]. Several effects of a low-protein (LP) diet can be provided with various advantages: (1) a decrease in nitrogen emission by feces, (2) low energy cost, (3) a reduction in the incidence of post-weaning diarrhea, and (4) alleviation of odor emission at the farm level [8]. In this regard, different levels of dietary CP in feed have been suggested by researchers [10,13,14,15] and the National Research Council (NRC), showing a difference in crude protein of 2–4% [16]. However, feeding a low-protein diet may induce problems, such as growth retardation and decreased meat quality [8]. Thus, the LP diet and the supplementation of crystalline amino acids are adapted to mitigate adverse effects on growth performance.

Comparisons of low CP and high CP diets have been conducted in several studies, but experiments with weaning to finishing pigs are scarce. We hypothesized that a 3% CP content reduction from the NRC’s (1998) recommendation would not affect growth performance, nutrient digestibility, or meat quality in weaning to finishing pigs. The objective of this study was to evaluate the optimal CP levels supplemented with the proper amount of crystalline AA based on the NRC guidelines from 1998 and 2012 [16,17], with a focus on assessing the impact on growth performance, fecal score, blood profiles, nutrient digestibility, and subsequent carcass characteristics of pigs from the weaning to the finishing stages.

## 2. Materials and Methods

### 2.1. Experimental Animals and Management

A total of 160 weaning pigs ([Yorkshire × Landrace] × Duroc) with an average body weight (BW) of 7.86 ± 1.05 kg were used for a 22-week feeding trial at the experimental farm of Seoul National University. All pigs were housed in an environmentally controlled building with a plastic-slotted floor facility (1.95 × 1.42 m) during weaning periods and a half-slotted concrete floor facility (2.60 × 2.84 m) during growing to finishing periods. Each pen had a feeder and a nipple drinker to provide ad libitum access.

### 2.2. Experimental Design and Diets

Following the NRC guidelines from 1998 and 2012, four dietary formulations were established in this experiment (Table 1, Table 2 and Table 3): high-protein (HP), medium-high-protein (MHP), medium-low-protein (MLP), and low-protein (LP). The HP diet was formulated based on the NRC guideline from 1998, and the MHP diet was formulated by decreasing the CP content of the HP diet by 1%. Furthermore, the MLP diet comprised a CP content reduced by 1.5% compared to the HP diet. The LP diet was formulated based on the NRC guideline from 2012, which recommends a lower protein content than the NRC guideline from 1998. Consistent metabolizable energy was provided in the growing–finishing periods to meet energy requirements and to manipulate dietary components without the intervention of energy level. In the diets of weaning pigs, the primary ingredients were ground corn, soybean meal (SBM), wheat, soy peptide, and lactose, and these ingredients were changed following weaning phases 1 (0–2 weeks) and 2 (3–6 weeks). Diets for growing–finishing pigs contained mainly ground corn, SBM, wheat, and palm kernel meal (PKM), and the composition of the diets was changed by growth stage (growing phases 1 and 2 at 7–10 weeks and 11–14 weeks and finishing phases 1 and 2 at 15–18 weeks and 19–22 weeks, respectively). Feed and water were available ad libitum, and the amount of feed was adjusted according to the BW. The residue of feed was recorded daily.

### 2.3. Growth Performance and Fecal Score

Body weight and feed intake were recorded at 0, 2, 6, 10, 14, 18, and 22 weeks to calculate the average daily gain (ADG), average daily feed intake (ADFI), and gain-to-feed ratio (G:F ratio).

Fecal score was measured at 8:00 am every day for 42 days during the weaning phase. Data were recorded by each pen and divided into 2 phases to assess the general pattern (Phase 1 and Phase 2). Fecal consistency scoring was based on the following index described by Sherman, et al. [18]: 0, normal (feces firm and well-formed); 1, soft consistency (feces soft and formed); 2, mild diarrhea (fluid feces, usually yellowish); and 3, severe diarrhea (feces watery and projectile). After recording the data, evidence of watery diarrhea was cleaned away every time to avoid infection from the previous day.

### 2.4. Blood Sampling and Analysis

Blood samples were taken from the jugular vein of 4 randomly selected pigs in each treatment to measure blood urea nitrogen (BUN), glucose, triglyceride, free fatty acids, and creatinine when the body weights were recorded. Collected blood samples were quickly centrifuged for 15 min by 3000 rpm at 4 °C. Then, the serum samples were transferred to 1.5 mL plastic tubes by pipette and stored at –20 °C until later analysis. The blood urea nitrogen (BUN), glucose, triglyceride, free fatty acids, and creatinine concentration were analyzed using a blood analyzer (Cobas 8000, Roche, Germany).

### 2.5. Digestibility Trial

A total of 12 crossbred barrows, averaging 32.08 ± 1.773 kg body weight, were allotted to individual metabolic crates in a completely randomized design (CRD) with 3 replicates to evaluate nutrient digestibility and nitrogen retention. Experimental diets of each growing phase (HP: 18% crude protein, MHP: 17% crude protein, MLP: 16.5% crude protein, LP: 15.68% crude protein) were provided to each treatment animals. A total collection method was used for the apparent nutrient digestibility. After a 5-day adaptation period, a 5-day collection period followed. To determine the first and last day of the collection days, 5% of ferric oxide and chromium oxide were added in the first and last experimental diet as selection markers, respectively. During the experimental period, water was provided ad libitum and all pigs were fed a daily level of 1.6 times the estimated maintenance requirement for energy (i.e., 106 kcal of ME per kg of BW^0.75^; NRC, 1998). Total urine was collected daily in a plastic container containing 50 mL of 4 N H_2_SO_4_ to avoid nitrogen evaporation, and containers were frozen during the 5-day collection period for nitrogen retention analysis. Collected feces and urine samples were stored at –20 °C until analysis.

### 2.6. Pork Quality and Carcass Characteristics

At the end of the experiment, four pigs were chosen for this study, including two female pigs (gilts) and two male pigs that had been castrated. These pigs were selected from the first repetition within a larger set of four repetitions, each representing different groups. At approximately 22 weeks of age, the pigs were humanely slaughtered at a commercial abattoir when they reached an approximate live weight of 105 kg. The analysis was performed by means of duplicated samples. Longissimus muscles were collected from nearby the 10th rib on the right side of the carcass. Because of the chilling procedure, 30 min after slaughter was regarded as the initial time. The pH and meat color were measured at 0, 3, 6, 12, and 24 h after the initial time. The pH was determined by a pH meter (Model, Thermo Orion, Waltham, Massachusetts, USA) and the meat color was determined by CIE color L*, a*, and b* value using a CR300 (Minolta Camera Co., Tokyo, Japan). Proximates of pork samples were analyzed by the method of AOAC (2005) [19].

The centrifuge method was used for measuring the water holding capacity (WHC) of the pork [20]. Longissimus muscle samples were grounded and sampled in a filter tube, heated in a water bath at 80 °C for 20 min, and then centrifuged for 10 min at 2000 rpm and 10 °C (Eppendorf centrifuge 5810 R, Taufkirchen, Germany). In order to calculate the cooking loss, longissimus muscles were packed with a polyethylene bag and heated in a water bath until the core temperature reached 72 °C. These were weighed before and after cooking. After heating, the samples were cored (0.5 inch in diameter) parallel to the muscle fiber, and the cores were used to measure the shear force (Warner-Bratzler Shear, Manhattan, USA). The cooking loss, shear force, and WHC of the pork were analyzed by animal origin food science, Seoul National University.

### 2.7. Fatty Acid Composition

Lipids in pork samples were extracted from duplicate 10 g samples with chloroform (methanol (2.1, *v*/*v*) [21] and shaking incubator (25 °C, 120 rpm)) for 24 h. Extracted lipids were filtered with filter paper (Whatman^TM^ No.4, Buckinghamshire, UK). Then, 25 mL of 0.88% NaCl was added to the filtered sample and centrifuged at 2090× *g* for 10 min (Continent 512 R, Hanil Co., Ltd., Incheon, Korea). The supernatant was separated, and we concentrated the pork lipids using N2 gas at 45 °C. After concentrating the lipids, 0.1 g was weighed into a 15 mL tube with 1 mL of internal standard (1 mg of undecanoic acid in 1 mL of iso-octane) and 1.5 mL of 0.5 N methanolic NaOH. The samples were heated in the water bath at 85 °C for 10 min and cooled to room temperature. After cooling, 2 mL of 14% BF3-methanol was added, and then we repeated the heating process one more time. After that, 2 mL of iso-octane and 1 mL of saturated NaCl was added and centrifuged at 2500 rpm for 3 min (Continent 512 R, Hanil Co., Ltd., Incheon, Korea). The upper layer containing fatty acid methyl ester (FAME) was dehydrated with anhydrous sodium sulfate and transferred to a vial. We analyzed the vial using a gas chromatograph (HP 7890, Agilent Technologies, Santa Clara, CA, USA) with a split ratio (50:1). A capillary column (DB-23, 60 m × 250 μm × 0.25 μm, Agilent, Santa Clara, CA, USA) was used. The injector and detector temperatures were maintained at 250 °C and 280 °C, respectively. The column oven temperature was as follows: 50 °C for 1 min, increased to 130 °C at 25 °C/min, 170 °C at 8 °C/min, and then held at 215 °C at 1.5 °C/min. Nitrogen was used as a carrier gas at a linear flow of 4 mL/min. Individual FAME was identified by comparison of the relative retention times of peaks from the samples, with those of the external standards (37 FAME mix and CLA mix, Supelco, Bellefonte, PA, USA) calculated based on the Korean Food Standards Codex.

### 2.8. Chemical and Statistical Analyses

The diets were ground by a Cyclotec 1093 Sample Mill (Foss Tecator, Hillerod, Denmark) and ground diets were analyzed. Collected excreta were pooled and dried in an air-forced drying oven at 60 °C for 72 h and ground into 1 mm particles in a Wiley mill for chemical analysis. Experimental diet and excreta samples were analyzed for content of dry matter (procedure 930.15; [22]), ash (procedure 942.05; [22]), ether extract (procedure 920.39; [22]) and N by using the Kjeldahl procedure with Kjeltec (Kjeltec TM 2200, Foss Tecator, Hilleroed, Denmark). Experimental diet and excreta samples were analyzed for CP content (nitrogen × 6.25; procedure 988.05; [22]).

The experimental data were analyzed as a randomized complete block design using the general linear model (GLM) procedure of SAS (ver. 9.4. SAS Inst. Inc., Cary, NC, USA). A pen was considered as an experimental unit for growth performance; for fecal score, a pen was considered as an experimental unit; and an individual pig was used as a unit for blood profile, nutrient digestibility, and pork quality for analysis. The differences were declared significant at *p* < 0.05 or highly significant at *p* < 0.01.

## 3. Results

### 3.1. Growth Performance

The effect of different protein levels on the growth performance of weaning to finishing pigs is presented in Table 4. No significant difference was found in BW. The average daily gain (ADG) of pigs fed the MLP diet during weeks 7–10 was higher than in other diets, and the ADG in the LP diet was lower compared to other treatments. The ADG in pigs fed the other two diets was positioned between the MLP and LP diets (*p* < 0.05). In weeks 15–22, the ADG of pigs fed the LP diet was higher than in other treatments, whereas the HP diet was lower in comparison to the other diets, and the ADG in pigs fed the MHP and MLP diets was positioned between the HP and LP diets (*p* < 0.05). There was no significant difference in ADFI. The G:F ratio of pigs fed the MLP diet was higher than that of pigs fed the other diets, and that of pigs fed the HP diet was lower than that of the other treatments, with the other two diets positioned between the HP and MLP diets in weeks 7–10 (*p* < 0.05).

### 3.2. The Diarrhea Score in Weaning Pigs and Nutrient Digestibility in Growing Pigs

A significant difference was not found in the apparent total tract digestibility (ATTD) of growing pigs or the fecal score of weaning pigs (Table 5 and Table 6).

### 3.3. Blood Profiles in Weaning to Finishing Pigs

The different levels of protein did not influence the concentrations of creatinine, free fatty acid, glucose, or triglyceride (Table 7). However, different protein levels affected the BUN concentration at 2, 6, and 18 weeks (*p* < 0.05); the BUN of pigs fed the LP diet showed the lowest value, whereas the highest value was found in pigs fed the HP diet, and the other two diets were similar to the HP diet or were positioned between pigs fed the HP and LP diets.

### 3.4. Meat Quality and Hunter Value in Longissimus Muscle after Slaughter

As shown in Table 8, protein, moisture, WHC, cooking loss, and pH were not influenced by the crude protein level in the diets. The hunter value L at 3 and 24 h after slaughter, and the hunter value b at 3 h after slaughter in pigs fed the LP diet, were highest, whereas they were the lowest in pigs fed the HP diet (Table 9: *p* < 0.05).

### 3.5. Fatty Acid Composition of Longissimus Muscle

The fatty acid composition of the longissimus muscle after slaughter is presented in Table 10. C14:0 and C16:0 in the longissimus muscle of pigs fed the MLP diet were higher than in other treatments (*p* < 0.05). C18:1, C20:1, and monounsaturated fatty acids (MUFA) in the longissimus muscle in pigs fed the MLP diet varied significantly (MLP > LP > HP > MHP; *p* < 0.05).

## 4. Discussion

Due to the emergence of various environmental issues related to swine farms and industries, numerous studies have been conducted to reduce the levels of crude protein and amino acids in pig diets with the aim of minimizing environmental impacts and promoting sustainable farming practices. However, the results from experiments have been variable and not constant according to the level of CP in pigs [8,23]. In addition, a low CP level may decrease pork quality and feed efficiency while increasing the carcass fat level [24]. In the current study, reduced CP levels did not show any detrimental effects on growth performance or pork quality. Specifically, different levels of CP in weaning diets (22% to 21% in phase 1 and 24.3% to 21% in phase 2) did not influence growth performance in agreement with previous studies [13,25]. Regardless of the dietary CP level, protein absorption after weaning can be limited in the small intestine due to villus atrophy, which induces reduced activity of intestinal enzymes and nutrient transporters [23,26]. The nutrients and protein that are not absorbed by the small intestine lead to an increase in protein fermentation in the cecum and colon. This can result in postweaning diarrhea [27]. The optimal level of AAs in an LP diet increases the abundance of healthy microbiota, such as *Prevotellaceae* and *Roseburia* bacteria, leading to improved growth performance and gut health [28]. Furthermore, a number of researchers have posited that there exists a positive correlation between the incidence rate of postweaning diarrhea and the level of CP content in the weaning pig diet [3]. As a result, the growth rate of weaning pigs can be readily affected by the level of CP. Based on our findings, the pigs that were fed an HP diet exhibited a numerically elevated diarrhea score in comparison to those fed alternative diets, but this did not influence growth performance of weaning pigs among the treatments. This is probably because the pigs fed an HP diet had more loose feces, but they did not have severe diarrhea, and the CP difference among the diets was lower than in previous experiments, as mentioned above [13,25]. Consequently, it can be inferred that an LP diet does not have a deleterious impact on growth performance, showing more solid feces among weaning pigs.

In contrast to the growth performance of weaning pigs, the different CP levels influenced that of growing–finishing pigs. The impairment of growth performance was not observed when the CP level was reduced by 2 to 4% from the NRC guideline from 1988 [29], and decreasing the CP level by 3 to 4% from the NRC guideline from 1988 in the current study showed similar results considering final BW. Moreover, synthetic AAs, such as L-lysine-HCl, DL-methionine, L-threonine, and tryptophan, were supplemented to increase nitrogen utilization and to reduce negative effects deriving from a reduction of CP. However, the response from varying CP levels was different in growing and finishing phases, respectively. In the growing phase, the ADG was decreased in pigs fed the LP diet based on the protein recommendation of the NRC (2012), except almost complete compensatory growth found in finishing pigs. Several factors, including feed restriction, early age growth retardation, and intestinal diseases, can potentially influence compensatory growth [30,31,32]. Rao, et al. [33] reported that lysine-restricted feeding in the early phase of finishing pigs decreases growth performance, but recovery was found, showing the magnitude of compensatory growth in the late phase. In our experiment, although the provision of lysine and methionine levels was similar regardless of the experimental diets, the ADG difference was found according to the growth stage. The experimental diets did not result in significant differences in final BW or ADFI over the entire study period. This compensatory growth might be derived from the reduced CP level with an increase in villus height reported by Limbach, et al. [34], who indicated that a reduced CP level from 21 to 19% in the weaning pig diet increased villus height. Similarly, villus height and intestinal surface area in humans are increased in response to dietary protein deficiency during lactation [35]. In addition, villus height and crypt depth ratio were increased in pigs fed LP diets, improving nutrient digestion and absorption in the small intestine [36]. In general, villus height is positively correlated with nutrient absorption and digestibility in pigs [37], and villus height is not related to the BW or age of pigs [38]. However, the growth rate of pigs can be differentiated by their age and BW. The growth rate in growing pigs is greater than in finishing pigs [39]. The abovementioned findings suggest that even if there is more absorption of protein in pigs fed the LP diet, the entire protein provision in the LP diet was not enough in the growing phase, but compensatory growth occurred along with increased protein absorption due to the relatively decreased growth rate in the finishing phase. Thus, the pigs fed the LP diet might have lasting efficacy for longer villus height in the small intestine by the finishing phase in comparison to the pigs fed higher protein diets, thereby leading to improved ADG in the finishing phase. It is therefore likely that the reduced CP level did not show a negative impact on growth performance due to the increasing villus height and nutrient absorption of pigs.

During the growing phase of the pigs, there was no impact of reduced CP level on ATTD of nutrient and nitrogen retention. This is probably because the 3% difference of CP level from the HP and LP diets did not influence digestibility or nitrogen retention, and nitrogen loss with microbiota from protein fermentation may hinder the precise calculation of nitrogen digestibility and retention. Instead, the blood urea nitrogen of pigs fed the LP diet at 2, 6, and 18 weeks was lower than in other treatments, which reflects the reduced CP level in the diet in accordance with previous studies [13,14,30].

The influence of different levels of CP on pork quality is still debatable. However, current findings showed that varying CP levels did not influence WHC, cooking loss, shear force (i.e., tenderness), or pH in line with previous findings that different levels of CP do not impact the pH or WHC of pork after slaughter [8,40,41]. Furthermore, Goodband, et al. [42] reported that increased lysine content in finishing pigs’ diet could decrease drip loss percentage in the experiment. However, the lysine level was similar in our experimental diets; thus, no difference in cooking loss was found regardless of dietary CP level. Hunter value L at 3 and 24 h, and hunter value b at 3 h after slaughter for pigs fed the LP diet, were higher than in the other diets, in agreement with previous reports [41,43]. Hunter value L reflects the level of light or dark, hunter value b is related to yellowness and blueness, and higher values of L and b are more desirable for meat [8]. Meat color can be affected by the proportion of unsaturated fatty acids (UFA) and saturated fatty acids (SFA). In particular, the effect of UFA on the reduction of meat color is more significant than that of SFA because UFAs are more susceptible to oxidation [2]. Furthermore, oxidation can alter the chemical structure of heme in myoglobin, thereby affecting meat color [44]. In the current study, the MUFA in the longissimus muscle of pigs fed MLP and LP diets was higher compared to the other treatments, whereas there was no difference in UFA, suggesting that reduced protein content in the diet could improve meat color. Furthermore, oleic acid (C18:1 n-9) was increased in the longissimus muscle of pigs fed MLP and LP diets. Utama, et al. [45] reported that the quality grade of beef could be increased in accordance with increasing the oleic acid proportion, which can improve eating quality for the consumer [15]. Furthermore, Grela, et al. [46] reported that a different proportion of major ingredients in feed may influence meat quality and the health-promoting value of meat due to different ratios of fatty acids. However, in this experiment, the difference in corn level in growing diets was 2–6%, and in finishing diets it was 2–7%. This difference in major ingredients did not influence meat color, in agreement with previous results. The results therefore suggest that reduced CP in the pig diet improves meat quality.

## 5. Conclusions

This study found that varying the level of CP, supplemented with increasing crystalline AA, did not influence the growth performance or incidence of diarrhea in weaning pigs. During the growing phase, however, the ADG was decreased in pigs fed the LP diet, but almost complete compensatory growth was found in finishing pigs, probably due to increasing villus height, which improved nutrient absorption in the finishing phase. Although varying CP levels did not affect WHC, cooking loss, shear force, or pH, meat color and oleic acid were improved in the longissimus muscle of pigs fed MLP and LP diets. Therefore, it can be inferred that a reduction in CP level with proper supplementation of crystalline AA during the weaning-to-finishing period for pigs does not exert significant adverse effects on either growth performance or meat quality.

## Figures and Tables

**Table 1 animals-13-01938-t001:** Ingredients and chemical compositions of the experimental diets in weaning pigs (Phase 1: 0–2 weeks; Phase 2: 3–6 weeks).

	Treatments ^1^							
	HP		MHP		MLP		LP	
	Phase 1	Phase 2	Phase 1	Phase 2	Phase 1	Phase 2	Phase 1	Phase 2
Ingredient, %								
Ground corn	29.95	42.97	32.18	45.58	33.45	46.76	37.32	48.13
SBM, 45%	38.6	30.1	35.4	26.9	33.8	25.3	28.6	23.7
Soy peptide	4.0	4.0	4.0	4.0	4.0	4.0	4.0	4.0
Wheat	11.8	13.4	12.3	13.5	12.5	13.8	13.1	13.8
Whey protein	4.0	2.0	4.0	2.0	4.0	2.0	4.0	2.0
Lactose	8.0	4.0	8.0	4.0	8.0	4.0	8.0	4.0
Soy oil	0.5	0.4	0.5	0.4	0.5	0.4	0.4	0.4
MDCP	1.3	1.1	1.3	1.2	1.3	1.2	1.4	1.3
Limestone	1.1	1.0	1.1	1.0	1.1	1.0	1.1	1.0
DL-methionine, 80%	0.1	0.1	0.1	0.1	0.1	0.1	0.2	0.1
Vit. Mix ^2^	0.1	0.1	0.1	0.1	0.1	0.1	0.1	0.1
Min. Mix ^3^	0.1	0.1	0.1	0.1	0.1	0.1	0.1	0.1
Salt	0.3	0.3	0.3	0.3	0.3	0.3	0.3	0.3
L-threonine, 99%	0.0	0.1	0.1	0.1	0.1	0.1	0.2	0.2
L-lysine-HCl, 78%	0.2	0.3	0.3	0.4	0.4	0.5	0.5	0.6
Tryptophan	0.0	0.0	0.2	0.2	0.3	0.3	0.6	0.4
Chemical composition								
ME, kcal/kg ^4^	3265.0	3265.0	3265.0	3265.0	3265.0	3265.0	3265.1	3265.0
Crude protein, % ^4^	23.7	20.9	22.7	19.9	22.2	19.4	20.6	18.9
Crude protein, % ^5^	22.1	24.3	21.5	20.9	21.1	21.0	21.4	21.0
Crude fat, % ^5^	1.0	1.6	1.1	1.5	1.3	2.2	1.4	1.3
Crude ash, % ^5^	6.2	5.8	7.1	5.8	6.3	4.5	7.3	5.6
Lysine, % ^4^	1.5	1.4	1.5	1.4	1.5	1.4	1.5	1.4
Methionine, % ^4^	0.4	0.4	0.4	0.4	0.4	0.4	0.4	0.4
Ca, % ^4^	0.8	0.7	0.8	0.7	0.8	0.7	0.8	0.7
P, % ^4^	0.7	0.6	0.7	0.6	0.7	0.6	0.7	0.6

^1^ HP: corn–SBM diet with NRC (1998) protein requirement, MHP: basal diet with NRC (1998) protein requirement—1%, MLP: basal diet with NRC (1998) protein requirement—1.5%, LP: basal diet with NRC (2012) protein requirement. ^2^ Provided the following quantities of vitamins per kg of complete diet: vitamin A, 8000 IU; vitamin D3, 1800 IU; vitamin E, 60 IU; thiamine, 2 mg; riboflavin, 7 mg; calcium pantothenic acid, 25 mg; niacin, 27 mg; pyridoxine, 3 mg; biotin, 0.2 mg; folic acid, 1 mg; vitamin B12, 0.03 mg. ^3^ Provided the following quantities of minerals per kg of complete diet: Se, 0.3 mg; I, 1 mg; Mn, 51.6 mg; CuSO_4_, 105 mg; Fe, 150 mg; Zn, 72 mg; Co, 0.5 mg. ^4^ Calculated values. ^5^ Analyzed value.

**Table 2 animals-13-01938-t002:** Ingredients and chemical compositions of the experimental diets in growing pigs (Phase 1: 7–10 weeks; Phase 2: 11–14 weeks).

	Treatments ^1^							
	HP		MHP		MLP		LP	
	Phase 1	Phase 2	Phase 1	Phase 2	Phase 1	Phase 2	Phase 1	Phase 2
Ingredient, %								
Ground corn	54.0	59.0	56.7	61.5	58.1	62.9	60.2	65.6
SBM, 45%	27.3	22.9	24.2	19.8	22.7	18.3	20.2	15.1
Wheat	10.0	10.0	10.0	10.0	10.0	10.0	10.0	10.0
Palm kernel meal	4.0	4.0	4.0	4.1	4.0	4.1	4.1	4.1
Tallow	1.8	1.6	1.8	1.6	1.8	1.6	1.8	1.6
MDCP	1.0	0.9	1.1	1.0	1.1	1.0	1.2	1.1
Limestone	1.0	0.9	1.0	0.9	1.0	0.9	1.0	0.9
DL-methionine, 80%	0.0	0.0	0.1	0.0	0.1	0.0	0.1	0.1
Vit. Mix ^2^	0.1	0.1	0.1	0.1	0.1	0.1	0.1	0.1
Min. Mix ^3^	0.1	0.1	0.1	0.1	0.1	0.1	0.1	0.1
Salt	0.3	0.3	0.3	0.3	0.3	0.3	0.3	0.3
L-threonine, 99%	0.0	0.0	0.1	0.1	0.1	0.1	0.1	0.1
L-lysine-HCl, 78%	0.2	0.2	0.3	0.3	0.4	0.3	0.5	0.5
Tryptophan	0.0	0.0	0.1	0.1	0.2	0.2	0.3	0.4
β-mannanase	0.1	0.1	0.1	0.1	0.1	0.1	0.1	0.1
Chemical composition								
ME, kcal/kg ^4^	3265.0	3265.0	3265.1	3265.1	3265.0	3265.1	3265.0	3265.1
Crude protein, % ^4^	18.0	16.3	17.0	15.3	16.5	14.8	15.7	13.8
Crude protein, % ^5^	18.1	15.8	16.1	15.0	16.6	14.6	15.3	14.3
Crude fat, % ^5^	3.2	4.3	3.7	5.2	4.2	4.6	3.6	5.5
Crude ash, % ^5^	6.2	6.2	7.1	5.8	7.3	5.8	6.3	5.6
Lysine, % ^4^	1.1	1.0	1.1	1.0	1.1	1.0	1.1	1.0
Methionine, % ^4^	0.3	0.3	0.3	0.3	0.3	0.3	0.3	0.3
Ca, % ^4^	0.7	0.6	0.7	0.6	0.7	0.6	0.7	0.6
P, % ^4^	0.6	0.5	0.6	0.5	0.6	0.5	0.6	0.5

^1^ HP: corn–SBM diet with NRC (1998) protein requirement, MHP: basal diet with NRC (1998) protein requirement—1%, MLP: basal diet with NRC (1998) protein requirement—1.5%, LP: basal diet with NRC (2012) protein requirement. ^2^ Provided the following quantities of vitamins per kg of complete diet: vitamin A, 8000 IU; vitamin D3, 1800 IU; vitamin E, 80 IU; thiamine, 2 mg; riboflavin, 7 mg; calcium pantothenic acid, 30 mg; niacin, 30 mg; pyridoxine, 3 mg; biotin, 0.2 mg; folic acid, 1 mg; vitamin B12, 0.10 mg. ^3^ Provided the following quantities of minerals per kg of complete diet: Se, 0.15 mg; I, 0.5 mg; Mn, 25.80 mg; CuSO_4_, 52.50 mg; Fe, 75 mg; Zn, 36 mg; Co, 0.25 mg. ^4^ Calculated values. ^5^ Analyzed value.

**Table 3 animals-13-01938-t003:** Ingredients and chemical compositions of the experimental diets in finishing pigs (Phase 1: 15–18 weeks; Phase 2: 19–22 weeks).

	Treatments ^1^							
	HP		MHP		MLP		LP	
	Phase 1	Phase 2	Phase 1	Phase 2	Phase 1	Phase 2	Phase 1	Phase 2
Ingredient, %								
Ground corn	59.0	67.9	61.5	70.6	62.9	71.9	65.6	75.2
SBM, 45%	22.9	14.7	19.8	11.6	18.3	10.1	15.1	6.2
Wheat	10.0	10.0	10.0	10.0	10.0	10.0	10.0	10.0
Palm kernel meal	4.0	4.1	4.1	4.1	4.1	4.2	4.1	4.2
Tallow	1.6	1.3	1.6	1.3	1.6	1.3	1.6	1.3
MDCP	0.9	0.6	1.0	0.7	1.0	0.7	1.1	0.8
Limestone	0.9	0.7	0.9	0.7	0.9	0.7	0.9	0.7
DL-methionine, 80%	0.0	0.0	0.0	0.0	0.0	0.0	0.1	0.0
Vit. Mix ^2^	0.1	0.1	0.1	0.1	0.1	0.1	0.1	0.1
Min. Mix ^3^	0.1	0.1	0.1	0.1	0.1	0.1	0.1	0.1
Salt	0.3	0.3	0.3	0.3	0.3	0.3	0.3	0.3
L-threonine, 99%	0.0	0.0	0.1	0.1	0.1	0.1	0.1	0.1
L-lysine-HCl, 78%	0.2	0.1	0.3	0.2	0.3	0.3	0.5	0.4
Tryptophan	0.0	0.0	0.1	0.2	0.2	0.2	0.4	0.5
β-mannanase	0.1	0.1	0.1	0.1	0.1	0.1	0.1	0.1
Chemical composition								
ME, kcal/kg ^4^	3265.0	3265.0	3265.1	3265.0	3265.0	3265.0	3265.0	3265.0
Crude protein, % ^4^	15.5	13.2	14.5	12.2	14.0	11.7	12.1	10.4
Crude protein, % ^5^	15.7	14.5	15.8	13.6	15.0	12.6	12.8	11.1
Crude fat, % ^5^	3.5	4.8	4.6	5.3	4.6	5.1	4.6	5.4
Crude ash, % ^5^	4.5	3.8	4.4	3.8	4.9	5.0	4.0	4.0
Lysine, % ^4^	0.8	0.7	0.8	0.7	0.8	0.7	0.8	0.7
Methionine, % ^4^	0.3	0.2	0.3	0.2	0.3	0.2	0.3	0.2
Ca, % ^4^	0.5	0.5	0.5	0.5	0.5	0.5	0.5	0.5
P, % ^4^	0.5	0.4	0.5	0.4	0.5	0.4	0.5	0.4

^1^ HP: corn–SBM diet with NRC (1998) protein requirement, MHP: basal diet with NRC (1998) protein requirement—1%, MLP: basal diet with NRC (1998) protein requirement—1.5%, LP: basal diet with NRC (2012) protein requirement. ^2^ Provided the following quantities of vitamins per kg of complete diet: vitamin A, 8000 IU; vitamin D3, 1800 IU; vitamin E, 80 IU; thiamine, 2 mg; riboflavin, 7 mg; calcium pantothenic acid, 30 mg; niacin, 30 mg; pyridoxine, 3 mg; biotin, 0.2 mg; folic acid, 1 mg; vitamin B12, 0.10 mg. ^3^ Provided the following quantities of minerals per kg of complete diet: Se, 0.15 mg; I, 0.5 mg; Mn, 25.80 mg; CuSO_4_, 52.50 mg; Fe, 75 mg; Zn, 36 mg; Co, 0.25 mg. ^4^ Calculated values. ^5^ Analyzed value.

**Table 4 animals-13-01938-t004:** Influence of different crude protein levels on growth performance in weaning to finishing pigs.

Criteria	Treatments ^1^				SEM ^2^	*p*-Value
	HP	HP	MLP	LP		
Body weight, kg						
Initial	7.86	7.86	7.86	7.86	0.001	0.662
2 weeks	9.57	9.61	9.76	9.76	0.182	0.826
6 weeks	19.46	19.26	19.11	18.49	0.647	0.745
10 weeks	37.44	37.23	37.94	34.32	0.889	0.067
14 weeks	57.85	57.90	57.63	53.97	1.212	0.125
18 weeks	78.63	78.66	79.62	76.89	1.711	0.728
22 weeks	102.49	103.88	105.1	102.38	1.560	0.583
Average daily gain, g						
0–2 weeks	123	125	136	136	13.0	0.830
3–6 weeks	354	345	334	312	21.7	0.586
0–6 weeks	276	272	268	253	15.4	0.744
7–10 weeks	642 ^ab^	642 ^ab^	673 ^a^	565 ^b^	20.3	0.024
11–14 weeks	729	638	703	702	34.4	0.832
7–14 weeks	686	690	688	633	20.9	0.235
15–18 weeks	770	769	814	849	38.1	0.424
19–22 weeks	823	870	879	879	17.7	0.142
15–22 weeks	797 ^b^	821 ^ab^	848 ^ab^	865 ^a^	12.5	0.018
Overall	614	624	632	614	12.9	0.597
Average daily feed intake, g						
0–2 weeks	320	286	306	295	19.8	0.653
3–6 weeks	737	769	723	710	41.1	0.765
0–6 weeks	598	608	584	572	28.5	0.813
7–10 weeks	1292	1263	1194	1113	46.1	0.087
11–14 weeks	1882	1835	1757	1735	71.8	0.474
7–14 weeks	1587	1549	1475	1424	56.5	0.240
15–18 weeks	2325	2180	2242	2294	90.3	0.695
19–22 weeks	2823	2780	2856	2775	86.4	0.896
15–22 weeks	2583	2490	2560	2543	75.6	0.845
Overall	1679	1635	1627	1598	44.7	0.652
Gain: feed ratio						
0–2 weeks	0.38	0.44	0.43	0.45	0.025	0.280
3–6 weeks	0.48	0.45	0.46	0.44	0.026	0.716
0–6 weeks	0.46	0.45	0.46	0.44	0.024	0.920
7–10 weeks	0.50 ^b^	0.51 ^ab^	0.57 ^a^	0.51 ^ab^	0.015	0.035
11–14 weeks	0.39	0.41	0.40	0.41	0.022	0.921
7–14 weeks	0.43	0.45	0.47	0.45	0.015	0.422
15–18 weeks	0.33	0.35	0.36	0.37	0.014	0.284
19–22 weeks	0.29	0.31	0.31	0.32	0.012	0.559
15–22 weeks	0.31	0.33	0.33	0.34	0.008	0.145
Overall	0.37	0.38	0.39	0.38	0.006	0.096

^1^ HP: corn–SBM diet with NRC (1998) protein requirement, MHP: basal diet with NRC (1998) protein requirement—1%, MLP: basal diet with NRC (1998) protein requirement—1.5%, LP: basal diet with NRC (2012) protein requirement. ^2^ Standard error of mean. ^a,b^ Means with different superscripts in the same row significantly differ (*p* < 0.05).

**Table 5 animals-13-01938-t005:** Influence of different crude protein levels on fecal score in weaning pigs.

Criteria	Treatments ^1^				SEM ^2^	*p*-Value
	HP	MHP	MLP	LP		
Fecal consistency ^3^						
0–2 weeks	1.27	1.29	1.21	1.13	0.162	0.220
3–6 weeks	1.98	1.35	1.36	1.33	0.265	0.302
0–6 weeks	1.71	1.33	1.30	1.14	0.209	0.327

^1^ HP: corn–SBM diet with NRC (1998) protein requirement, MHP: basal diet with NRC (1998) protein requirement—1%, MLP: basal diet with NRC (1998) protein requirement—1.5%, LP: basal diet with NRC (2012) protein requirement. ^2^ Standard error of mean. ^3^ Fecal consistency scoring index: 0, normal (feces firm and well-formed); 1, soft consistency (feces soft and formed); 2, mild diarrhea (fluid feces, usually yellowish); and 3, severe diarrhea (feces watery and projectile).

**Table 6 animals-13-01938-t006:** Influence of different crude protein levels on nutrient digestibility in growing pigs ^1^.

Criteria	Treatment ^2^				SEM ^3^	*p-*Value
	HP	MHP	MLP	LP		
Nutrient digestibility, %						
Dry matter	87.9	90.7	83.3	86.3	1.079	0.330
Crude protein	84.6	87.4	77.4	81.3	1.493	0.225
Crude ash	68.9	72.4	50.0	58.7	3.516	0.670
Crude fat	81.2	81.6	69.3	77.6	2.195	0.580
Nitrogen retention, g/d						
N intake	80.0	76.5	75.3	74.2	0.652	-
Fecal N	12.3	9.7	17.0	13.9	1.104	0.230
Urinary N	16.4	13.1	16.6	13.8	0.759	0.190
Total N excretion	28.7	22.8	33.7	27.6	1.672	0.302
N retention ^4^	51.3	53.7	41.7	46.6	1.836	0.453
N retention, % ^5^	64.1	70.2	55.3	62.8	2.234	0.428

^1^ Least squares means for 3 pigs per treatment. Initial BW: 32.08 kg. ^2^ HP: corn–SBM diet with NRC (1998) protein requirement, MHP: basal diet with NRC (1998) protein requirement—1%, MLP: basal diet with NRC (1998) protein requirement—1.5%, LP: basal diet with NRC (2012) protein requirement. ^3^ Standard error of mean. ^4^ N retention = N intake—Fecal N—Urinary N. ^5^ N retention (%) = N retention/N intake × 100.

**Table 7 animals-13-01938-t007:** Influence of different crude protein levels on blood profiles of weaning to finishing pigs ^1^.

Criteria	Treatment ^2^				SEM ^3^	*p-*Value
	HP	MHP	MLP	LP		
Blood urea nitrogen, mg/dL						
2 weeks	19.6 ^A^	16.2 ^AB^	14.8 ^AB^	11.4 ^B^	1.11	0.004
6 weeks	18.8 ^A^	16.4 ^A^	16.4 ^A^	11.4 ^B^	1.03	0.004
10 weeks	12.8	14.1	9.1	8.2	0.96	0.065
14 weeks	11.2	9.2	10.0	7.8	0.57	0.234
18 weeks	17.2 ^A^	10.7 ^B^	10.4 ^BC^	6.3 ^C^	1.16	0.002
22 weeks	9.7	8.4	8.1	4.6	0.75	0.146
Creatinine, mg/dL						
2 weeks	0.7	0.8	0.7	0.7	0.07	0.638
6 weeks	0.8	0.8	0.8	0.7	0.05	0.374
10 weeks	0.9	1.0	1.0	0.9	0.04	0.275
14 weeks	1.2	1.2	1.3	1.2	0.03	0.513
18 weeks	1.2	1.2	1.3	1.3	0.03	0.588
22 weeks	1.4	1.3	1.3	1.2	0.04	0.353
Free fatty acid, μEq/L						
2 weeks	74.0	87.8	49.5	63.0	21.68	0.653
6 weeks	80.3	87.0	79.0	63.5	25.17	0.923
10 weeks	92.5	88.7	88.5	70.0	6.81	0.666
14 weeks	167.8	84.8	64.8	114.0	12.23	0.519
18 weeks	85.0	155.5	92.3	90.0	11.15	0.107
22 weeks	121.0	195.8	64.0	134.8	11.92	0.076
Glucose, mg/dL						
2 weeks	93.0	85.3	93.8	92.3	5.13	0.640
6 weeks	94.8	84.8	87.3	91.5	4.68	0.477
10 weeks	92.8	90.5	87.8	99.3	2.34	0.358
14 weeks	84.3	91.8	92.8	85.3	1.49	0.123
18 weeks	88.0	82.8	91.8	90.0	1.90	0.485
22 weeks	89.5	92.3	93.5	87.8	2.01	0.786
Triglyceride, mg/mL						
2 weeks	40.2	30.8	29.5	35.5	5.32	0.385
6 weeks	42.5	31.3	32.3	35.8	6.68	0.642
10 weeks	58.5	63.0	60.5	66.3	4.68	0.950
14 weeks	60.5	83.0	66.0	61.8	5.51	0.500
18 weeks	75.3	61.3	61.8	62.8	5.80	0.842
22 weeks	57.0	55.3	69.3	81.0	7.46	0.248

^1^ Least squares means for 4 pigs per treatment. ^2^ HP: corn–SBM diet with NRC (1998) protein requirement, MHP: basal diet with NRC (1998) protein requirement—1%, MLP: basal diet with NRC (1998) protein requirement—1.5%, LP: basal diet with NRC (2012) protein requirement. ^3^ Standard error of mean. ^ABC^ Means in a same row with different superscript letters significantly differ (*p* < 0.01).

**Table 8 animals-13-01938-t008:** Influence of different crude protein levels on meat quality of longissimus muscle after slaughter ^1^.

Criteria	Treatments ^2^				SEM ^3^	*p-*Value
	HP	MHP	MLP	LP		
Proximate analysis						
Protein (%)	24.03	23.99	23.59	23.77	0.226	0.440
Moisture (%)	72.06	72.02	72.29	72.50	0.459	0.867
WHC	71.93	72.98	73.32	70.87	1.614	0.708
Cooking loss	30.12	30.40	32.40	30.68	1.119	0.505
Shear force	56.30	57.70	57.26	53.71	6.155	0.966
pH, time after slaughter						
0 h	6.07	5.97	5.91	5.75	0.069	0.808
3 h	5.78	5.70	5.62	5.58	0.049	0.560
6 h	5.59	5.55	5.56	5.43	0.034	0.480
12 h	5.63	5.51	5.61	5.50	0.035	0.655
24 h	5.69	5.58	5.60	5.50	0.032	0.590

^1^ Least squares means for 4 pigs per treatment. ^2^ HP: corn–SBM diet with NRC (1998) protein requirement, MHP: basal diet with NRC (1998) protein requirement—1%, MLP: basal diet with NRC (1998) protein requirement—1.5%, LP: basal diet with NRC (2012) protein requirement. ^3^ Standard error of mean.

**Table 9 animals-13-01938-t009:** Influence of different crude protein levels on meat color of longissimus muscle after slaughter ^1^.

Criteria	Treatments ^2^				SEM ^3^	*p-*Value
	HP	MHP	MLP	LP		
Hunter value, L ^4^						
0 h	37.69	39.70	40.87	43.16	0.814	0.075
3 h	38.45 ^b^	41.80 ^ab^	42.73 ^ab^	45.99 ^a^	0.921	0.038
6 h	40.48	42.76	43.70	46.76	0.833	0.125
12 h	42.58	44.73	45.67	48.10	0.723	0.212
24 h	43.37 ^b^	46.45 ^ab^	45.76 ^b^	49.22 ^a^	0.777	0.023
Hunter value, a* ^5^						
0 h	9.22	8.85	9.36	9.18	0.329	0.690
3 h	10.18	9.55	11.05	10.51	0.352	0.752
6 h	10.62	10.16	11.16	10.86	0.351	0.658
12 h	12.06	11.66	12.03	12.04	0.376	0.568
24 h	11.06	11.17	11.42	11.39	0.355	0.540
Hunter value, b* ^6^						
0 h	4.70	4.84	4.97	5.49	0.146	0.244
3 h	4.99 ^b^	5.33 ^b^	5.69 ^ab^	6.25 ^a^	0.162	0.035
6 h	5.32	5.65	5.88	6.39	0.170	0.125
12 h	5.69	5.87	6.02	6.44	0.146	0.097
24 h	6.06	5.95	5.93	6.42	0.137	0.239

^1^ Least squares means for 4 pigs per treatment. ^2^ HP: corn–SBM diet with NRC (1998) protein requirement, MHP: basal diet with NRC (1998) protein requirement—1%, MLP: basal diet with NRC (1998) protein requirement—1.5%, LP: basal diet with NRC (2012) protein requirement. ^3^ Standard error of mean. ^4^ L—luminance or brightness (vary from black to white). ^5^ a*—red·green component (+a = red, −a = green). ^6^ b*—yellow·blue component (+b = yellow, −b = blue). ^a,b^ Means with different superscripts in the same row significantly differ (*p* < 0.05).

**Table 10 animals-13-01938-t010:** Influence of different crude protein levels on fatty acid composition of longissimus muscle ^1^.

Criteria	Treatments ^2^				SEM ^3^	*p-*Value
	HP	MHP	MLP	LP		
Fatty acid composition, mg/g						
C14:0	1.29 ^b^	1.30 ^b^	2.17 ^a^	1.59 ^b^	0.120	0.010
C16:0	36.99	36.64	50.13	47.51	2.253	0.061
C16:1	3.43 ^b^	3.26 ^b^	5.98 ^a^	4.08 ^b^	0.348	0.005
C17:0	1.40	1.01	1.35	1.74	0.091	0.055
C18:0	22.40	21.88	27.15	26.97	1.343	0.644
C18:1 n-9	45.20 ^bc^	41.62 ^c^	70.28 ^a^	58.65 ^ab^	3.712	0.021
C18:2 n-6	45.68	45.41	44.64	51.51	2.682	0.789
C18:3 n-6	0.56	0.54	0.56	0.72	0.037	0.430
C18:3 n-3	0.80	0.71	0.77	0.79	0.051	0.858
C20:1	0.71 ^bc^	0.59 ^c^	1.08 ^a^	0.91 ^ab^	0.062	0.021
C20:2	2.00	1.74	1.82	2.49	0.135	0.212
C21:0	2.77	2.74	2.62	3.45	0.175	0.391
C20:4 n-6	20.11	17.78	17.04	21.39	1.324	0.487
C20:5 n-3	0.70	0.56	0.55	0.76	0.046	0.198
C22:6 n-3	0.51	0.46	0.44	0.59	0.047	0.597
SFA ^4^	64.85	63.56	83.43	81.25	3.786	0.375
UFA ^5^	119.69	112.66	143.14	141.88	6.382	0.446
MUFA ^6^	49.34 ^bc^	45.47 ^c^	77.35 ^a^	63.64 ^ab^	4.085	0.017
PUFA ^7^	70.35	67.19	65.80	78.24	4.203	0.664
UFA/SFA ratio	1.84	1.76	1.72	1.75	0.020	0.328

^1^ Least squares means for 4 pigs per treatment. ^2^ HP: corn–SBM diet with NRC (1998) protein requirement, MHP: basal diet with NRC (1998) protein requirement—1%, MLP: basal diet with NRC (1998) protein requirement—1.5%, LP: basal diet with NRC (2012) protein requirement. ^3^ Standard error of mean. ^a,b,c^ Means with different superscripts in the same row significantly differ (*p* < 0.05). ^4^ SFA = saturated fatty acids. ^5^ UFA = unsaturated fatty acids. ^6^ MUFA = monounsaturated fatty acids. ^7^ PUFA = polyunsaturated fatty acids.

## Data Availability

Not applicable.

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
