# Peer review of "The Effect of Reduced Crude Protein on Growth Performance, Nutrient Digestibility, and Meat Quality in Weaning to Finishing Pigs"

_animals, 2023, doi:10.3390/ani13121938_

Round 1

Reviewer 1 Report

The protein levels in pig diets were studied from weaning to finishing.

The authors should be clear about the current polynomial orthogonal contrasts. Every period, the difference between treatments should be the same. Please explain clearly.

The digestibility trial did not represent the overall digestibility of diets from weaning to finishing. Because the authors only conducted one digestibility trial (at about 32 kg). Please think about taking it out.

The authors studied carcass characteristics and meat quality in only four pigs per treatment out of 40 pigs per treatment. What were the criteria for selecting pigs for slaughter? It is obvious that the variation in data on carcass characteristics and meat quality was high, making it difficult to determine whether the results were due to imposed treatments.

Author Response

Dear Reviewer 1,

Thank you for your insightful comments and valuable suggestions regarding our manuscript. We highly appreciate your thorough review and have carefully considered all of your feedback. Please find attached our response letter as a Word file, addressing each of your concerns and providing further clarification on certain points. We hope that our revisions adequately address your concerns and we thank you once again for your time and effort in reviewing our work.

Best regards,

Geonil Lee

Reviewer 2 Report

The manuscript: "The effect of reduced crude protein on growth performance, nutrient digestibility, and meat quality in weaning to finishing pigs" presents a current topic. The nutritional in rational feeding  needs should be covered, but at the same time an excess of protein in the feed should be avoided. Among other things, reducing the protein in the diet decreases the excretion of nitrogen into the environment. A low-protein diet, without damage to the achieved optimal body weight gains, is possible provided that the limiting amino acids are supplemented by adding them in their pure form.

The authors use a high level of maize in the feed mixtures, which, in my opinion, could have influenced the obtained result. It is a pity that only some feed ingredients have been analytically determined (protein, fat, ash), and the rest of chemical composition - calculated. In my opinion, it is necessary to obtain amino acid determinations. Especially since on line 280 the Authors write:  "In our experiment, although the provision of lysine and methionine level was similar regardless of experimental diets". It should be verified by performing chemical analyses.

I also missed the determination of the chemical composition of meat and the level of protein and even the composition of amino acids of meat.

English was not reviewed.

All the best and stay safe

Author Response

(The authors gave the same response as above.)

Reviewer 3 Report

The manuscript investigates the effect of varying levels of dietary CP, supplemented with increasing proportions of crystalline AA, on growth performance, carcass characteristics, total tract nutrient digestibility and some general health markers in wean-to-finish pigs. The authors conclude that there was little negative impact of reducing dietary CP on pig growth or the other parameters measured. It would be useful to include a true concluding statement, something related to the value of this work, or implications of this work on commercial pig production. Word requirements for the abstract may limit this but considering there is really a lot of work in this area of the lack of negative effect of reduced CP diet on pig growth, there does not appear to be anything really new from this work. Again, notwithstanding limitations in space in the abstract, some indication of how ‘reduced’ the low CP diet was or how much crystalline AA was needed to be added would be helpful for the reader to put the work into context. The overall objective was to investigate impacts of lower dietary CP, the problem is that in the first 2 phases, there are minimal differences in formulated CP content, with the exception of Phase 2 (3-6 weeks) but even then, the HP diet had a CP content greater than the phase 1 in that same treatment, and the other diets has the same CP content. By week 7 – 14 there were at least formulated differences in CP content although the differences are so small that within the MHP and MLP differences were easily in the realm of measurement error in mixing diets. This is also true in the finisher diets. This study seems to really be a comparison between NRC 1998 and 2012 diet CP recommendations as opposed to an investigation of increasingly lower CP diets. The growth performance data supports this where the differences noted are really between

Other specific comments

-        L32-33. This sentence is confusing. Appears the point is that at some point after week 10 pigs on LP diet displayed compensatory gain?

-        L99-102 talk about ‘growth phase 1 and 2’ and ‘growing phases 1 and 2’ based on the weeks these are different but the use of almost the same names is a bit confusing.

-        Table 4. Superscripts are used for some significant linear responses but not all? See comment re statistics as to whether superscripts is most appropriate in the first plae.

-        L155  ”using as alter” what is this?

-        Statistics: treatments were not equally spaced yet L195 indicates that was an assumption of the analysis. The space between HP, MHP, MLP were described as equal but the LP was a greater proportion lower than the others (at least after the first 42 days). Typically there is not superscripts on individual treatments when linear/quadratic analysis has been completed. Rather, where a quadratic effect is noted, determination of a ‘breakpoint’ would be suitable to determine at what point does the direction of the linear response change. Discussion with statistician well versed in livestock studies would help to determine best method of analysis and hence, reporting. In the results section (eg. L202-214) it appears both an ANOVA and a linear/quadratic statistical model was used. Choose one that is most suitable, not both.

-        L252-253 these values do not match with values reported in the diet formulation table for the nursery phase, particularly for phase 2.

-        L264-265. This may be true but overall the fecal scores did not reflect diarrhea or loose stool was a concern in this trial so the statement (L265-267) over interprets the data.

-        L333 similarly over extrapolated the data to make this statement.

Some locations where statements are unclear or confusing.

Author Response

Dear Reviewer,

Thank you for your insightful comments and valuable suggestions regarding our manuscript. We highly appreciate your thorough review and have carefully considered all of your feedback. Please find attached our response letter as a Word file, addressing each of your concerns and providing further clarification on certain points. We hope that our revisions adequately address your concerns and we thank you once again for your time and effort in reviewing our work.

Best regards,

Geonil Lee

Round 2

Reviewer 1 Report

Because low protein diets may not affect growth performance, particularly meat quality, the title should be changed. Dietary ingredients appear to have a greater potential to differentiate meat quality.

Why was the energy in diets the same from growing to finishing?

Meat quality, in my opinion, differed due to feed ingredients in diets; for example, high corn meal in diets may result in a different fatty acid profile, particularly high oleic acid, as well as meat color, Hunter value, L, in meat than low corn meal in diets. Line 325 needs to be changed.

When you chose a pig for slaughter at 105 kg at 22 weeks, this criteria did not represent the treatment effect because you only chose the top of the pig in each treatment. Pigs in each group had average final weights less than 105 kg. Please explain your reasoning in detail.

Please include the P-value in tables 4-10.

Author Response

Dear Reviewer

We express our sincere appreciation for your insightful comments and suggestions, which have proven to be invaluable in expanding our understanding of the realm of meat science. As a result of your guidance, we have broadened our perspectives and enhanced our knowledge in this field.

Please find attached the response letter in the form of a Word file.

With great respect,

Geonil Lee
